# Prognostic value of postoperative serum carcinoembryonic antigen levels in colorectal cancer patients who smoke

Chih-Sheng Huang[1,2]*, Chin-Yau Chen[1,2], Li-Kuo Huang[2,3], Wei-Shu Wang[2,4], Shung-Haur Yang[1,2,5]

**1** Department of Surgery, National Yang-Ming University Hospital, Yilan, Taiwan, **2** School of Medicine, National Yang-Ming University, Taipei, Taiwan, **3** Department of Radiology, National Yang-Ming University Hospital, Yilan, Taiwan, **4** Division of Oncology, Department of Internal Medicine, National Yang-Ming University Hospital, Yilan, Taiwan, **5** Division of Colon and Rectal Surgery, Department of Surgery, Taipei Veterans General Hospital, Taipei, Taiwan

\* imalive20@hotmail.com

**Data Availability Statement:** All relevant data are within the paper and its Supporting Information files.

## Abstract

Serum carcinoembryonic antigen (CEA) levels can help predict the prognosis of colorectal cancer patients. Accordingly, high preoperative CEA levels that is not restored after surgery are indicative of a worse outcome. On the other hand, smoking can increase serum CEA levels independently of the disease status. Thus, we aimed to evaluate the impact of smoking on the prognostic value of serum CEA levels. This retrospective cohort study included 273 patients who underwent curative resection for stage I–III colorectal adenocarcinoma at a single institution, between January 2010 and December 2017. Patients were grouped as follows: group A, normal preoperative and postoperative CEA levels (n = 152); group B, elevated preoperative CEA levels that returned to reference values after surgery (n = 69); and group C, elevated postoperative serum CEA levels (n = 52). Patients were also grouped according to their smoking history: group S (current smokers, n = 79) and group NS (never and former smokers, n = 194). Group A showed a higher 3-year disease-free survival (DFS) rate (84.9%) than groups B (75.4%) and C (62.0%) ($p < 0.001$). Postoperative serum CEA levels were significantly higher in the S group than in the NS group (2.6 vs. 3.1 ng/mL, $p = 0.009$), whereas preoperative levels were similar (3.8 vs. 4.1, $p = 0.182$). Further, smokers showed higher 3 year-DFS rates than nonsmokers in group C (83.3% vs. 43.9%, $p = 0.029$). This suggests that while elevated postoperative CEA levels are associated with lower DFS rates in never and former smokers, they are not associated with lower DFS rates in current smokers. We conclude that persistent smoking alters the prognostic value of postoperative serum CEA levels in colorectal cancer patients and that, consequently, alternative surveillance strategies need to be developed for colon cancer patients with smoking habits.

**Funding:** National Yang-Ming University Hospital The funders had no role in study design, data collection and analysis, decision to publish, or preparation of the manuscript.

**Competing interests:** The authors have declared that no competing interests exist.

## Introduction

Serum carcinoembryonic antigen (CEA) is the most widely used tumor marker for patients with colorectal cancer. Most published guidelines, including those from the National Comprehensive Cancer Network and the American Society of Oncologists, recommend postoperative serum CEA testing every 3–6 months [1,2]. Preoperative CEA levels ≥5.0 ng/mL have been reported to adversely impact survival, independently of tumor stage [3–5]. In 2000, the Colorectal Working Group of the American Joint Committee on Cancer (AJCC) recommended modifications of the TNM staging system to differentiate between tumors of patients with normal vs. elevated serum CEA levels at presentation [6]. However, serum CEA levels are no longer included as a factor for staging in the AJCC TNM staging system 8th edition.

High preoperative serum CEA levels do not return to reference values after surgery in approximately one-third of the patients with colorectal cancer. This indicates the presence of persistent disease and the need for further evaluation [7,8]. Therefore, consideration of both preoperative and postoperative serum CEA levels might effectively predict the prognosis of patients with colorectal cancer [9].

Although CEA is a recognized tumor marker, its levels can be influenced by many factors [10–12]. Tobacco use is one of the most common causes of CEA elevation [13–15], which can lead to inaccurate cancer diagnosis and prognosis. This study aimed to determine the effect of cigarette smoking on the prognostic value of serum CEA levels. We hypothesized that current smokers with elevated postoperative CEA levels might not have lower disease-free survival (DFS) rates.

## Materials and methods

After obtaining the institutional review board of National Yan-Ming University Hospital's approval (NYMUH IRB No.2020A001) and a waiver of the requirement for patient consent, prospectively maintained databases were queried for all consecutive studies.

### Patients

A total of 444 patients with colorectal adenocarcinomas received curative treatment at the National Yang Ming University Hospital between January 1, 2010 and December 31, 2017. We excluded 171 patients from the study because of stage IV disease (n = 52), loss to follow-up (n = 26), diagnosis of carcinoma *in situ* (n = 29), and incomplete CEA data (n = 64). Thus, our study ultimately comprised 273 patients.

Patients were grouped according to their CEA status as follows: group A, normal (<5.0 ng/ mL) pre- and postoperative serum CEA levels (n = 152); group B, elevated (≥5.0 ng/mL) preoperative but normal postoperative serum CEA levels (n = 69); and group C, elevated postoperative serum CEA levels (n = 52). The patients were also grouped according to their smoking history. Group S comprised current smokers, defined as people who have smoked in their lifetime and currently smoke cigarettes (n = 74). Group NS comprised never smokers (defined as people who have never smoked) and former smokers (defined as people who have smoked in their lifetime but had quit smoking before surgery) (n = 199, 173 never smokers and 26 former smokers).

### Data collection

We prospectively developed a computerized database at our hospital and updated it constantly. The recorded variables included the patients' demographic data and major comorbidities;

family history of cancer; tumor location, number, and stage; macro- and microscopic pathological characteristics; and patient status at the last follow-up.

## Evaluation and treatment

Tumor staging was based on the TNM system described in the 7th edition of the International Union Against Cancer/AJCC [16]. Serum CEA levels were measured in a single laboratory using an Elecsys E170 analyzer (Roche Diagnostics, Indianapolis, IN, USA), with a recommended upper reference limit of 5 ng/mL. Preoperative serum CEA levels were measured immediately before the surgery, and postoperative serum CEA levels were measured 4 to 6 weeks after surgery. All patients were evaluated via staging workups including a colonoscopy, complete blood count, serum CEA determination, chest radiography, and computed tomography (CT) of the abdomen.

All patients underwent radical surgical resection. Twenty-two patients also underwent preoperative neoadjuvant chemoradiation therapy (CRT) for locally advanced rectal cancer; preoperative CEA was determined before initiating CRT. The CR protocol was described in our previous study [17]. Postoperative adjuvant chemotherapy was considered for 104 patients with pathologic stage III disease. Of these, 12 did not receive adjuvant chemotherapy due to refusal or poor performance status. The chemotherapy regimens were 5-fluoruracil/leucovorin/oxaliplatin (FOLFOX) in 69 patients, capecitabine/oxaliplatin (XELOX) in 2 patients, oral tegafur/uracil (UFUR) in 19 patients, and oral capecitabine in 2 patients. Postoperative adjuvant chemotherapy was also administered to 80 patients with both pathologic stage II disease and other risk factors such as pathologic stage pT4, lymphovascular invasion, perineural invasion, and anastomosis leakage. The regimen was FOLFOX in 37 patients and oral UFUR in 43 patients.

## Surveillance protocol

All patients were followed-up in the outpatient department every 3 months in the first 2 years, every 6 months in the third and fourth year, and annually thereafter. The follow-up examinations included determination of serum CEA levels, chest and abdominopelvic CT, and colonoscopy. It is our policy to perform the first follow-up colonoscopy 6 months after surgery in patients in whom a complete colonoscopy study had not been or could not be performed before surgery. If the patient had undergone complete colonoscopy before surgery, the first follow-up colonoscopy was performed 1 year after surgery.

## Statistical analysis

Optimal cutoff CEA values were determined using receiver operating characteristic (ROC) curve analysis and Youden's index. Chi-square test, Mann-Whitney U test, and Kruskal-Wallis analyses were used to analyze categorical and continuous variables, respectively. Survival curves were plotted using the Kaplan-Meier method, and survival values were compared using the log-rank test. Death and disease recurrence were treated as events in the analysis. Differences in DFS rates in the univariate analysis were assessed using the log-rank test. Hazard ratios and associations with DFS were determined via multivariable Cox regression analysis. Variables with $p < 0.05$ on univariate analysis were included in the multivariable model. Data were analyzed using MedCalc statistical software version 19.0.3 (MedCalc Software bvba, Ostend, Belgium), and $p < 0.05$ was considered statistically significant.

## Results

### Patient characteristics

**Overall population.**   Out of the 273 patients included in our study, 149 (54.6%) were men. The median age was 71 years (range 28–93 years), and the median pre- and postoperative serum CEA concentrations were 3.9 ng/mL (range, 0.6–263.5 ng/mL) and 2.7 ng/mL (range 0.5–84.9 ng/mL), respectively. Tumors were located in the right colon in 99 patients (36.3%), in the left colon in 101 patients (37.0%), and in the rectum in 73 patients (26.7%).

The median follow-up interval was 46 months (range, 4–117 months). Tumors recurred in 58 patients (21.2%) before the last follow-up. The sites of tumor recurrence were the liver (n = 24), lungs (n = 22), peritoneum (n = 9), para-aortic lymph nodes (n = 9), bone (n = 4), and brain (n = 2). Local recurrence was observed in 13 patients. The 3-year and 5-year DFS rate for all patients was 78.0% and 69.1%, respectively.

**By CEA status.**   The demographic and clinical features of patients grouped according to their CEA status (group A: normal pre- and postoperative serum CEA levels, group B: elevated preoperative and normal postoperative serum CEA levels, and group C: elevated pre- and post-operative levels) are shown in Table 1. There were no significant differences in sex distribution, median age, tumor location, histologic differentiation, or lymphovascular and perineural invasion status among the three groups. Group B and group C patients tended to have a more advanced T and N stage than did group A patients. The percentage of current smokers was higher in group C (46%) than in group A (24%) and B (20%) ($p$ = 0.002).

**By smoking history status.**   The demographic and clinical features of patients grouped according to their smoking history status [smoking (S) and nonsmoking (NS) patients] are shown in Table 2. Compared with the NS group, the S group contained significantly more men (42% vs. 88%, $p$ < 0.001) and younger patients (median age, 73 vs. 68 years, $p$ = 0.009). Postoperative serum CEA levels were significantly higher in the S than in the NS group (2.6 vs. 3.1 ng/mL, $p$ = 0.009), whereas preoperative serum CEA levels were similar (3.8 vs. 4.1 ng/mL, $p$ = 0.182) between groups. More patients had elevated postoperative serum CEA levels in the S than in the NS group (32.4% vs. 14.1%, $p$ < 0.001). There were no significant differences in the pathological characteristics or the 3-year DFS rate between groups (80.2% vs. 77.1%, p = 0.485).

### Disease-free survival rates

The overall 3-year DFS rate was significantly higher in group A (84.2%), with respect to group B (74.3%) and C (62.0%) ($p$ = 0.001; Fig 1A). Among NS, the 3-year DFS rate was also significantly higher in group A with respect to group B and C (84.9% vs. 75.4% vs. 43.9%, $p$ < 0.001; Fig 1B). However, among S, differences in the 3 year-DFS rates among groups A, B, and C diminished (82.3% vs. 69.2% vs. 83.3%, $p$ = 0.772; Fig 1C).

The overall 3-year DFS rate was similar between smokers and nonsmokers (80.2% vs. 77.1%, $p$ = 0.485; Fig 2A). In the subgroup analysis, the 3-year DFS rate was also similar between smokers and nonsmokers in groups A and B (78.8% vs. 81.8%, $p$ = 0.853; Fig 2B). In contrast, smokers had higher 3 year-DFS rate than nonsmokers in group C (83.3% vs. 43.9%, $p$ = 0.029; Fig 2C).

Since the sample size of those on XELOX and capecitabine was too small for conclusions, we compared FOLFOX/XELOX versus UFUR/capecitabine instead. The analysis comparing between patients treated with different chemotherapy regimens showed no significant difference in DFS (FOLFOX/XELOX vs. UFUR/Capecitabine, 76.3% vs. 73.4%, $p$ = 0.506).

**Table 1. Clinical characteristics by CEA status.**

| | Group A (n = 152) | Group B (n = 69) | Group C (n = 52) | *p* |
|---|---|---|---|---|
| **Sex** | | | | |
| Male | 80 (53%) | 36 (52%) | 33 (53%) | 0.239 |
| Female | 72 (47%) | 33 (48%) | 19 (37%) | |
| **Age (years), median (range)** | 70 (32–93) | 69 (28–89) | 72 (53–88) | 0.493 |
| **Tumor location** | | | | |
| Right colon | 52 (34%) | 22 (32%) | 25 (48%) | 0.222 |
| Left colon | 59 (39%) | 24 (35%) | 18 (35%) | |
| Rectum | 41 (27%) | 23 (33%) | 9 (17%) | |
| **Preoperative CEA levels (ng/mL), median (range)** | 2.6 (0.6–4.9) | 8.6 (5.0–97.1) | 8.9 (2.8–263.5) | < 0.001 |
| **Postoperative CEA levels (ng/mL), median (range)** | 2.2 (0.5–4.9) | 2.6 (0.8–4.7) | 6.4 (5.1–84.9) | < 0.001 |
| **T Stage** | | | | |
| T1 | 22 (14%) | 2 (3%) | 2 (3%) | < 0.001 |
| T2 | 32 (21%) | 4 (6%) | 5 (10%) | |
| T3 | 94 (62%) | 60 (87%) | 39 (75%) | |
| T4 | 4 (3%) | 3 (4%) | 6 (12%) | |
| **N Stage** | | | | |
| N0 | 109 (72%) | 34 (49%) | 26 (50%) | 0.002 |
| N1 | 33 (22%) | 21 (31%) | 19 (37%) | |
| N2 | 10 (6%) | 14 (20%) | 7 (13%) | |
| **TNM stage** | | | | |
| I | 47 (31%) | 5 (7%) | 7 (13%) | < 0.001 |
| II | 62 (41%) | 29 (42%) | 19 (37%) | |
| III | 43 (28%) | 35 (51%) | 26 (50%) | |
| **Differentiation** | | | | |
| Well | 6 (4%) | 0 | 4 (8%) | 0.210 |
| Moderately | 142 (93%) | 68 (99%) | 46 (88%) | |
| Poorly | 4 (3%) | 1 (1%) | 2 (4%) | |
| **LVI** | | | | |
| Yes | 90 (59%) | 42 (61%) | 31 (60%) | 0.973 |
| No | 62 (41%) | 27 (39%) | 21 (40%) | |
| **PNI** | | | | |
| Yes | 18 (12%) | 11 (16%) | 6 (12%) | 0.667 |
| No | 134 (88%) | 58 (84%) | 46 (88%) | |
| **Chemotherapy regimen** | | | | |
| FOLFOX | 54 (36%) | 36 (52%) | 16 (31%) | < 0.001 |
| XELOX | 0 (0%) | 2 (3%) | 0 (0%) | |
| UFUR | 35 (23%) | 19 (28%) | 18 (34%) | |
| Capecitabine | 0 (0%) | 0 (0%) | 2 (4%) | |
| No | 63 (41%) | 12 (17%) | 16 (31%) | |
| **Smoking** | | | | |
| Yes | 36 (24%) | 14 (20%) | 24 (46%) | 0.002 |
| No | 116 (76%) | 55 (80%) | 28 (54%) | |

Data are presented as n (%), unless otherwise indicated.

PNI, perineural invasion; LVI, lymphovascular invasion; TNM, tumor-node-metastasis

**Table 2. Patient characteristics by smoking status.**

| | NS (n = 199) | S (n = 74) | *p* |
|---|---|---|---|
| **Sex** | | | |
| Male | 84 (42%) | 65 (88%) | < 0.001 |
| Female | 115 (58%) | 9 (12%) | |
| **Age (year), median (range)** | 73 (32–93) | 68 (28–87) | 0.009 |
| **Tumor location** | | | |
| Right colon | 81 (41%) | 18 (24%) | 0.038 |
| Left colon | 67 (34%) | 34 (46%) | |
| Rectum | 51 (25%) | 22 (30%) | |
| **Preoperative CEA levels (ng/mL), median (range)** | 3.8 (0.6–97.1) | 4.1 (1.2–263.5) | 0.182 |
| **Postoperative CEA levels (ng/mL), median (range)** | 2.6 (0.5–84.9) | 3.1 (0.5–12.5) | 0.009 |
| **T Stage** | | | |
| T1 | 20 (10%) | 6 (8%) | 0.912 |
| T2 | 31 (15%) | 10 (14%) | |
| T3 | 139 (70%) | 54 (73%) | |
| T4 | 9 (5%) | 4 (5%) | |
| **N Stage** | | | |
| N0 | 121 (61%) | 48 (65%) | 0.753 |
| N1 | 56 (28%) | 17 (23%) | |
| N2 | 22 (11%) | 9 (12%) | |
| **TNM stage** | | | |
| I | 45 (23%) | 14 (19%) | 0.502 |
| II | 76 (38%) | 34 (46%) | |
| III | 78 (39%) | 26 (35%) | |
| **Differentiation** | | | |
| Well | 6 (3%) | 4 (6%) | 0.490 |
| Moderately | 187 (94%) | 69 (93%) | |
| Poorly | 6 (3%) | 1 (1%) | |
| **LVI** | | | |
| Yes | 119 (60%) | 44 (59%) | 0.959 |
| No | 80 (40%) | 30 (41%) | |
| **PNI** | | | |
| Yes | 23 (12%) | 12 (16%) | 0.307 |
| No | 176 (88%) | 62 (84%) | |
| **Chemotherapy regimen** | | | |
| FOLFOX | 77 (39%) | 29 (39%) | 0.843 |
| XELOX | 2 (1%) | 0 (0%) | |
| UFUR | 52 (26%) | 20 (27%) | |
| Capecitabine | 2 (1%) | 0 (0%) | |
| No | 66 (33%) | 25 (34%) | |

Data are presented as n (%), unless otherwise indicated.

CEA, carcinoembryonic antigen; PNI, perineural invasion; LVI, lymphovascular invasion; TNM, tumor-node-metastasis

Univariate analysis showed that preoperative serum CEA levels, postoperative serum CEA levels, tumor stage, age, lymphovascular invasion, and perineural invasion status were predictive of DFS (Table 3). In a multivariable analysis, only postoperative serum CEA levels, tumor stage, and age were significant independent prognostic factors for DFS (Table 4).

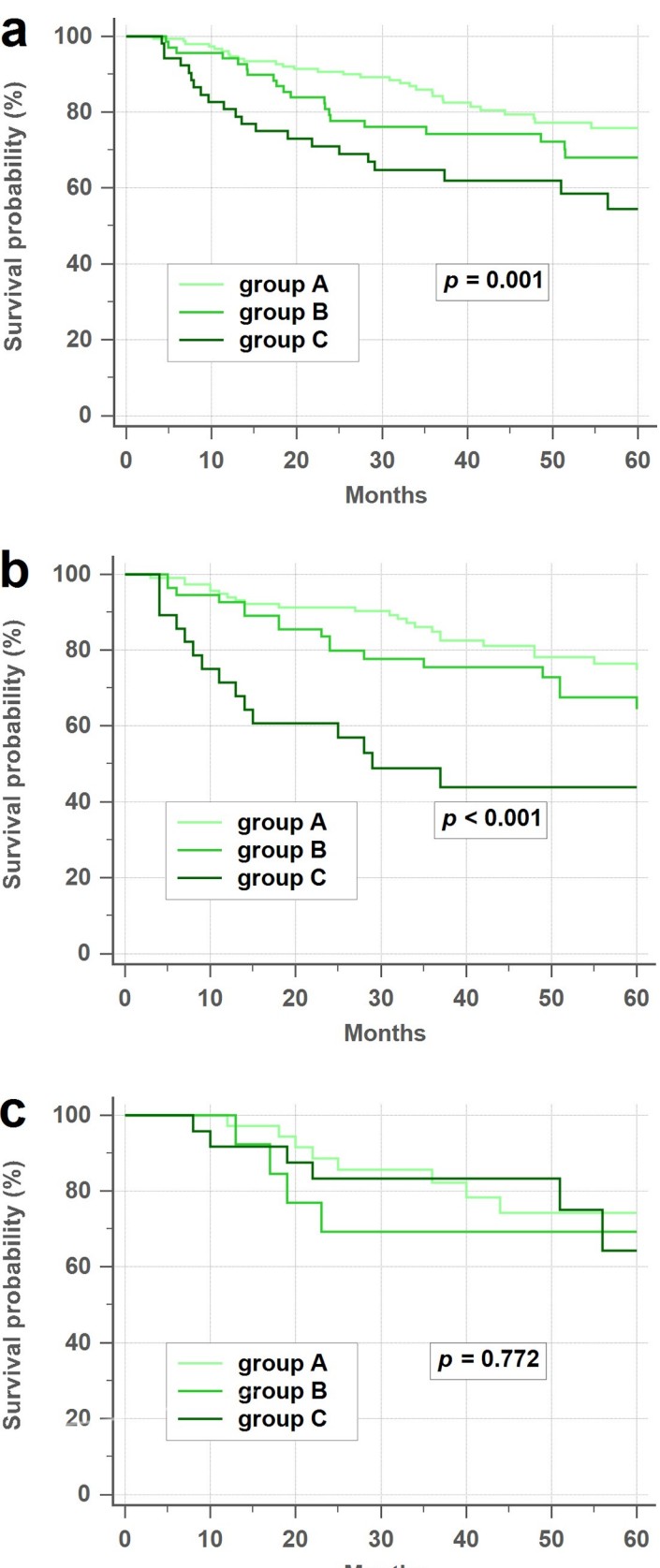

**Fig 1. Disease-free survival rates according to the CEA status.** (a) Overall disease-free survival (DFS) rates. (b) Never/former smoker DFS rates. (c) Current smoker DFS rates. CEA, carcinoembryonic antigen.

## Optimal cutoff values of serum CEA levels for smoking patients

The optimal CEA level cutoff values for smoking patients were determined using ROC curve analysis. Preoperative and postoperative serum CEA levels showed no prognostic efficacy for colorectal cancer in smoking patients, with areas under the receiver operating characteristic curve (AUC) values of 0.543 and 0.583, respectively. The optimal cutoff value for preoperative serum CEA levels was 2.9 ng/ml; this had a sensitivity, specificity, and Youden index of 78.9%, 34.6%, and 0.136, respectively. Meanwhile, the optimal cutoff value for postoperative serum CEA levels was 2.2 ng/ml; this had a sensitivity, specificity, and Youden index of 85.0%, 31.4%, and 0.165, respectively.

## Discussion

In this study, serum CEA level status significantly influenced the DFS rate of colon cancer patients. Patients with normal pre- and postoperative serum CEA levels (group A) had a better prognosis than did patients with elevated preoperative but normal postoperative CEA levels (group B). Patients with elevated postoperative serum CEA levels (group C) had the worst outcome. These results are consistent with those of previous studies [18,19]. However, as a novel finding, we show that current smokers with elevated postoperative CEA levels might not have a worse prognosis than non-smoking patients with similar postoperative CEA levels. To our best knowledge, this is the first study to determine the prognostic value of CEA status in smoking and nonsmoking patients with colorectal cancer.

Tobacco use is a well-known risk factor for serum CEA levels elevation [13–15]. Potential mechanisms by which smoking might increase CEA levels include chronic immune cell recruitment and inflammation [20]. Lung function and CEA levels significantly improve after 3 months of smoke cessation [21]. Thus, we placed former smokers (i.e., those who had quit smoking before or after surgery) in group NS along with never smokers, rather than in the S group, which consisted of patients who smoked both before and after surgery. Group S had higher postoperative CEA levels than did the NS group (2.6 vs. 3.1, $p = 0.009$), whereas both groups had similar preoperative levels (3.8 vs. 4.1, p = 0.182). This suggests that the contribution of cancer cells to serum CEA levels may be significantly higher than that of smoking in the preoperative period. Moreover, serum CEA values obtained in smoking patients may not reflect the actual levels of CEA produced by cancer cells, especially after surgery. This could explain why elevated postoperative CEA levels are less prognostic in current smokers.

Although smoking influences serum CEA concentrations, our data showed no significant differences in DFS between current smokers and former/never smokers (80.2% vs. 77.1%, p = 0.485). In our study, the smoker group included significantly younger patients (median age, 73 vs. 68 years, p = 0.009), which may have affected this result. The association between smoking status and survival in patients with colorectal cancer has yet to be established, with most [22–25] but not all studies [26–28], linking cigarette smoking to worse survival rates.

Serum CEA level is a widely accepted tumor marker, particularly for colorectal cancer, and its determination is standardized, inexpensive, and easily available. Elevated preoperative CEA levels are thought to be an independent prognostic factor in colorectal carcinoma. In previous studies, overall survival rates were lower when preoperative serum CEA levels were elevated, regardless of the disease stage [3,19,29]. However, some studies reported that postoperative serum CEA levels were better predictors than were preoperative serum CEA levels. In these

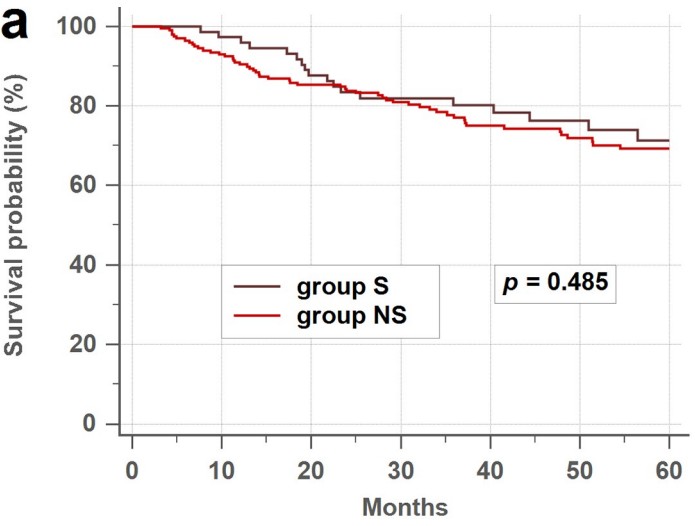

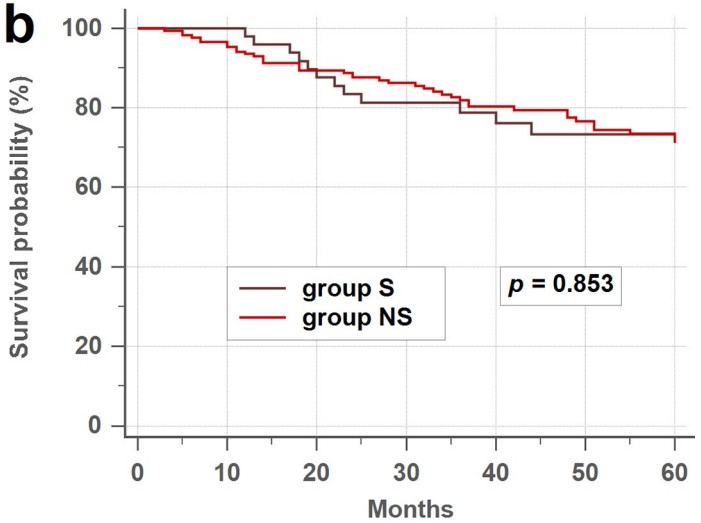

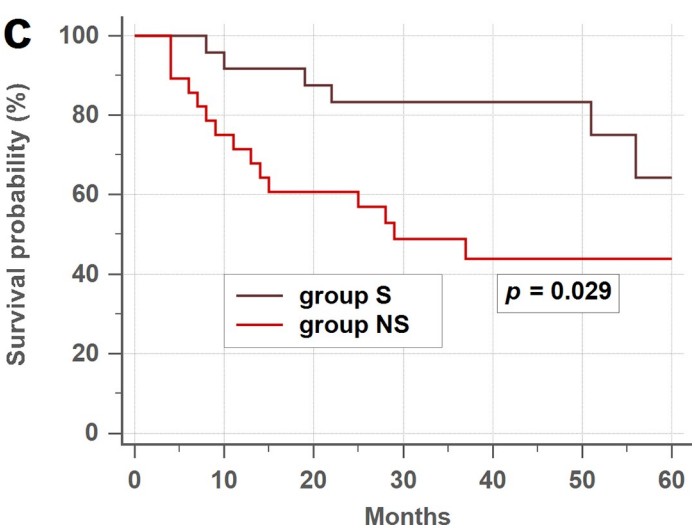

**Fig 2. Disease-free survival rates between smokers and nonsmokers.** (a) Overall disease-free survival (DFS) rates. (b) Patients with normal postoperative serum CEA levels. (c) Patients with elevated postoperative serum CEA levels. CEA, carcinoembryonic antigen.

studies, postoperative, but not preoperative, CEA status was a significant prognostic predictor in multivariable analyses [7,9,30]. In fact, current guidelines do not support the use of elevated preoperative CEA as an indicator for adjuvant chemotherapy [1,2]. Konishi el al. found no significant differences in the 3-year DFS rates between patients with normal postoperative serum CEA levels and those with normal preoperative CEA levels [9]. Elevated preoperative serum CEA levels are not informative when the postoperative levels are normal; thus, preoperative

**Table 3. Univariate analysis of prognostic factors for disease-free survival.**

|  | No. of patients | 3-year DFS rate | *p* |
|---|---|---|---|
| **Sex** |  |  |  |
| **Male** | 149 | 74.5% | 0.523 |
| **Female** | 124 | 81.0% |  |
| **Age, years** |  |  |  |
| **< 75** | 168 | 81.2% | 0.005 |
| **≥ 75** | 105 | 72.8% |  |
| **Preoperative CEA** |  |  |  |
| **< 5** | 152 | 84.2% | 0.002 |
| **≥ 5** | 121 | 70.1% |  |
| **Postoperative CEA** |  |  |  |
| **< 5** | 221 | 81.1% | < 0.001 |
| **≥ 5** | 52 | 62.0% |  |
| **Tumor location** |  |  |  |
| **Right** | 99 | 76.2% | 0.455 |
| **Left** | 101 | 74.0% |  |
| **Rectum** | 73 | 83.7% |  |
| **Differentiation** |  |  |  |
| **Well** | 10 | 67.5% | 0.710 |
| **Moderately** | 256 | 78.6% |  |
| **Poorly** | 7 | 71.4% |  |
| **LVI** |  |  |  |
| **Yes** | 163 | 71.9% | 0.023 |
| **No** | 110 | 85.9% |  |
| **PNI** |  |  |  |
| **Yes** | 35 | 60.4% | 0.010 |
| **No** | 238 | 80.5% |  |
| **TNM Stage** |  |  |  |
| **I** | 59 | 93.7% | < 0.001 |
| **II** | 110 | 84.4% |  |
| **III** | 104 | 62.3% |  |
| **Smoking** |  |  |  |
| **Yes** | 79 | 80.2% | 0.485 |
| **No** | 194 | 77.1% |  |

CEA, carcinoembryonic antigen; PNI, perineural invasion; LVI, lymphovascular invasion; TNM, tumor-node-metastasis

**Table 4. Multivariate analysis of prognostic factors for disease-free survival.**

|  | Hazard ratio | 95% CI | *p* |
|---|---|---|---|
| **Postoperative CEA** |  |  |  |
| < 5 | 1 |  |  |
| ≥ 5 | 1.814 | 1.016–3.241 | 0.044 |
| **TNM stage** |  |  |  |
| I | 1 |  |  |
| II | 2.085 | 0.797–5.457 | 0.135 |
| III | 4.603 | 1.791–11.828 | 0.002 |
| **Age, years** |  |  |  |
| < 75 | 1 |  |  |
| ≥ 75 | 1.643 | 1.057–2.553 | 0.028 |

CEA, carcinoembryonic antigen; TNM, tumor-node-metastasis

serum CEA level determination could be disregarded as a prognostic factor. In agreement, elevated postoperative serum CEA levels were a better predictor than elevated preoperative serum CEA levels in our multivariable model.

In our study, preoperative and postoperative serum CEA levels showed no prognostic efficacy for smoker patients, with AUC values close to 0.5 (0.543 and 0.583, respectively). It was not possible to redefine serum CEA level elevation by simply increasing the threshold because we found no linear correlation between serum CEA values and DFS in this subgroup. The optimal cutoff values for preoperative and postoperative serum CEA levels in smokers were, respectively, 2.9 ng/ml and 2.2 ng/ml, which were even lower than the global standard CEA cutoff value of 5.0 ng/ml. As such, we failed to identify a predictive threshold for smokers before and after surgery.

In addition to CEA status, we found that tumor stage and age were also independent prognostic factors for DFS on multivariable analysis. Regional lymph node involvement is one of the strongest predictors of outcome following surgical resection of colorectal cancers. Nodal spread, rather than elevated serum CEA concentrations, is an indication for adjuvant therapy for colorectal cancer in most guidelines [1,2]. Advanced age has been shown to reduce overall survival and DFS rates and, to a lesser extent, cancer-specific survival rates in patients with colorectal cancer [31–34].

The major limitation of our study is the variability of the adjuvant chemotherapy regimens. However, an analysis comparing the four chemotherapy regimens (XELOX, FOLFOX, oral UFUR, and oral capecitabine) showed no significant differences in DFS among them. As a further limitation, we quantified smoking only according to self-reports at a single time point before surgery, a method that is relatively unreliable. Additional limitations include the relatively small number of former smokers in the NS group, which likely resulted in limited statistical power, and the lack of consideration of factors (e.g., diabetes, liver disease, and acute or chronic inflammation) that can also generate confusing CEA results. Under consideration of cost, molecular tests were only reserved for patients with stage IV disease in our hospital. The patients included in this study lacked molecular profiles such as *KRAS*, *NRAS*, and *BRAF*. Lastly, our study was retrospective, with a relatively small number of patients, and the follow-up period for some patients was short. In the future, large prospective studies analyzing CEA kinetics via measurements of follow-up serum CEA levels are required. The amount and duration of smoking and type of cigarette should also be taken into consideration in future studies.

## Conclusion

Our findings demonstrate that persistent smoking can increase serum CEA levels in patients with colorectal cancer, affecting the postoperative prognostic value of serum CEA levels in current smokers. Elevated postoperative serum CEA level is associated with lower DFS rates in never and former smokers, but not in current smokers. Therefore, colorectal cancer patients that are current smokers may need the characterization of alternative tumor markers, useful as surveillance strategy.

## Supporting information

**S1 Raw Data.**
(XLSX)

## Acknowledgments

The authors would like to acknowledge Wei-Ching Lin for her help in coordinating the clinical aspects of this study.

## Author Contributions

**Conceptualization:** Chih-Sheng Huang.

**Data curation:** Chih-Sheng Huang, Wei-Shu Wang.

**Formal analysis:** Chih-Sheng Huang.

**Methodology:** Chih-Sheng Huang, Chin-Yau Chen.

**Project administration:** Li-Kuo Huang.

**Resources:** Chin-Yau Chen, Li-Kuo Huang.

**Software:** Chih-Sheng Huang.

**Supervision:** Chin-Yau Chen, Li-Kuo Huang, Wei-Shu Wang, Shung-Haur Yang.

**Visualization:** Li-Kuo Huang.

**Writing – original draft:** Chih-Sheng Huang.

**Writing – review & editing:** Shung-Haur Yang.

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
