## [Decision Letter · Decision Letter 0]

30 Mar 2020

PONE-D-20-04276

Prognostic value of serum carcinoembryonic antigen levels in colorectal cancer patients who smoke

PLOS ONE

Dear Dr. Huang,

Thank you for submitting your manuscript to PLOS ONE. After careful consideration, we feel that it has merit but does not fully meet PLOS ONE’s publication criteria as it currently stands. Therefore, we invite you to submit a revised version of the manuscript that addresses the points raised during the review process.

ACADEMIC EDITOR: The study is interesting and adds something new to this field. Please kindly respond to the valuable comments from the reviewers. 

We would appreciate receiving your revised manuscript by May 14 2020 11:59PM. To enhance the reproducibility of your results, we recommend that if applicable you deposit your laboratory protocols in protocols.io, where a protocol can be assigned its own identifier (DOI) such that it can be cited independently in the future. For instructions see: http://journals.plos.org/plosone/s/submission-guidelines#loc-laboratory-protocols

We look forward to receiving your revised manuscript.

Kind regards,

Jason Chia-Hsun Hsieh, M.D. Ph.D

Academic Editor

PLOS ONE

Journal Requirements:

2. In the ethics statement in the manuscript and in the online submission form, please provide additional information about the patient records used in your retrospective study, including: a) whether all data were fully anonymized before you accessed them and b) the date range (month and year) during which patients' medical records were accessed.

Reviewers' comments:

Reviewer's Responses to Questions

**Comments to the Author**

1. Is the manuscript technically sound, and do the data support the conclusions?

Reviewer #1: Yes

Reviewer #2: Partly

Reviewer #3: Yes

2. Has the statistical analysis been performed appropriately and rigorously? 

Reviewer #1: I Don't Know

Reviewer #2: No

Reviewer #3: Yes

3. Have the authors made all data underlying the findings in their manuscript fully available?

Reviewer #1: No

Reviewer #2: Yes

Reviewer #3: Yes

4. Is the manuscript presented in an intelligible fashion and written in standard English?

Reviewer #1: No

Reviewer #2: Yes

Reviewer #3: Yes

5. Review Comments to the Author

Reviewer #1: In this study, authors aimed to evaluate the impact of smoking on the prognostic value of serum CEA levels. In this single site, they retrospectively collected clinical data, treatment outcomes, smoking status and CEA level before and after operation. They found that although patients whose CEA levels were normal preoperative and postoperative showed the best 3-year DFS, elevated postoperative CEA levels were associated with lower DFS rates in never and former smokers, but were not associated with lower DFS rates in current smokers.

The strength of this study is that they found that persistent smoking altered the prognostic value of postoperative serum CEA levels in colorectal cancer patients. This is the first study to determine the prognostic value of CEA status in smoking and nonsmoking patients with colorectal cancer. Before accepted for publication, they are still some issues.

Major issue

1. Different stage will lead to different outcome. In colon cancer, molecular markers such as K-Ras,N-Ras and B-Raf are also an important factors. This study included stage I, II and III colon-rectal cancer. Authors should list the distribution of stages within different groups. If possible, analysis of molecular subtypes are also important.

2. In line 223, authors mentioned about groups C had the worse outcome. But in group C, there were more patients with right side tumors compared with Group B and A, and more patients were LN positive compared with Group A. In summery, group C was already with worse prognostic factors, nothing to do with CEA level before and after operation. Interesting, fewer patients in this group had received adjuvant chemotherapy compared with group B, which might also lead to worse outcome.

Minor issues

1. Authors may add regimen of different adjuvant chemotherapy in each group.

2. How can authors distinguish local recurrence from metachronous colon cancers (line 149)?

3. In line 280, authors mentioned an analysis comparing the four chemotherapy regimens showed no significant differences in DFS among them. Can you provide the reference?

4. In this study, authors should define current smoker and former smoker more accurately. You can see a paper which was published at J Clin Oncol. 2013 Jun 1;31(16):2016-23. Especially former smoker, some patients might just stop smoking before operation. Another issue is previous smoking might already lead to lung injury, which may also lead to elevation of CEA.

Reviewer #2: According to your statistic method, I don’t think it support your conclusion and the title. You should focus on the patients with persistent elevated CEA and using NS/S to compare disease-free survival if prognostic factors are balanced between two groups.

Reviewer #3: General Comments:

This is a generally well-written manuscript.

However, there are still some issues regarding scientific novelty, scientific interpretation of the results that need to be well-addressed.

Major comments:

1. In group C, the author defined elevated pre-operative and post-operative CEA group. However, in table1, the range of pre-operative CEA level is 2.8-263.5 ng/ml. Which means some patient’s CEA level was normal before operation.

The author should carefully defined the group C.

2. One important finding in this study is post-operative CEA level is significantly higher in smoking group than non-smoking group (2.6 vs. 3.1 ng/ml). Although the p value is significant, the difference is too small in clinical practice. My other question is in non-smoking group, the mean post-operative CEA level is 2.6 ng/ml, range from 0.5-84.9 ng/ml. However, in smoking group, the mean post-operative CEA level is 3.1 ng/ml, range from 0.5-12.5 ng/ml. It seems the distribution of CEA level in non-smoking group is very wide. Do the author have the confidence to claim the post-operative CEA level is really higher in smoking group than non-smoking group?

Minor comments:

In abstract, the post-operative CEA level in smoking group is 2.5 ng/ml. However, in the table 2, the level is 2.6 ng/ml.

6. PLOS authors have the option to publish the peer review history of their article (what does this mean?). If published, this will include your full peer review and any attached files.

Reviewer #1: No

Reviewer #2: No

Reviewer #3: No

---

## [Author Response · Author response to Decision Letter 0]

13 Apr 2020

Reviewer #1

Comments 1: 

Different stage will lead to different outcome. In colon cancer, molecular markers such as K-Ras,N-Ras and B-Raf are also an important factors. This study included stage I, II and III colon-rectal cancer. Authors should list the distribution of stages within different groups. If possible, analysis of molecular subtypes are also important.

Response 1:

Yes. We completely agree with the reviewer’s pertinent comments. The distribution of stages within different groups are now provided in the table below and in the revised manuscript (Results section, Table 1, page 9).

 Group A

(n = 152) Group B

(n = 69) Group C

(n = 52) p

TNM stage 

I 47 (31%) 5 (7%) 7 (13%) < 0.001

II 62 (41%) 29 (42%) 19 (37%) 

III 43 (28%) 35 (51%) 26 (50%) 

We agree that the molecular markers play an important role in colorectal cancer prognosis. However, under consideration of cost, molecular tests were only reserved for patients with stage IV disease in our hospital. The patients included in this study lacked molecular profiles such as KRAS, NRAS, and BRAF. We added this as a limitation in the revised manuscript (Discussion section, page 19, lines 295-297).

Comments 2: 

2. In line 223, authors mentioned about groups C had the worse outcome. But in group C, there were more patients with right side tumors compared with Group B and A, and more patients were LN positive compared with Group A. In summery, group C was already with worse prognostic factors, nothing to do with CEA level before and after operation. Interesting, fewer patients in this group had received adjuvant chemotherapy compared with group B, which might also lead to worse outcome.

Response 2:

Yes. We agree with the reviewer’s insightful concern. Tumor sides, tumor stage, and chemotherapy are all well-known prognostic factors of colorectal cancer. As shown in Fig 1a, we found that in the CEA group, the overall 3-year DFS rate was significantly higher in group A (84.2%), with respect to group B (74.3%) and C (62.0%). Univariate analysis in our study showed that the CEA group was predictive of disease-free survival.

Univariate analysis of the CEA group for disease-free survival

 No. of patients 3-year DFS rate p

CEA group 

Group A 152 84.2% 0.001

Group B 69 74.3% 

Group C 52 64.7% 

If we select the CEA group rather than the preoperative and postoperative CEA as a variable, CEA group, tumor stage, and age were significant independent prognostic factors for disease-free survival in multivariable analysis.

Multivariate analysis of prognostic factors for disease-free survival

 Hazard ratio 95% CI p

CEA group 

Group A 1 

Group B 1.299 0.760–2.222 0.338

Group C 2.356 1.369–4.060 0.002

TNM stage 

I 1 

II 2.085 0.797–5.457 0.135

III 4.603 1.791–11.828 0.002

Age, years 

< 75 1 

≥ 75 1.643 1.057–2.553 0.028

As a result, we believe that tumor side, stage, chemotherapy, and elevated postoperative CEA may all have contributed to the poorer prognosis in group C.

Comments 3: 

Authors may add regimen of different adjuvant chemotherapy in each group.

Response 3:

We thank the reviewer for the insightful comment. The regimen of different adjuvant chemotherapy within different groups are provided in the tables below and in the revised manuscript

Regimen of adjuvant chemotherapy within CEA groups 

 Group A

(n = 152) Group B

(n = 69) Group C

(n = 52) p

Regimen 

FOLFOX 54 (36%) 36 (52%) 16 (31%) < 0.001

XELOX 0 (0%) 2 (3%) 0 (0%) 

UFUR 35 (23%) 19 (28%) 18 (34%) 

Capecitabine 0 (0%) 0 (0%) 2 (4%) 

No 63 (41%) 12 (17%) 16 (31%) 

(Results section, Table 1, page 9)

Regimen of adjuvant chemotherapy within smoking groups

 NS (n = 199) S (n = 74) p

Regimen 

FOLFOX 77 (39%) 29 (39%) 0.843

XELOX 2 (1%) 0 (0%) 

UFUR 52 (26%) 20 (27%) 

Capecitabine 2 (1%) 0 (0%) 

No 66 (33%) 25 (34%) 

(Results section, Table 2, page 12)

Comments 4: 

How can authors distinguish local recurrence from metachronous colon cancers?

Response 4:

We thank the reviewer for the thoughtful comment. Local recurrence was observed in 13 patients in our study. Most of these cases initially had locally advanced cancer with adjacent organ involvement. The clinical features of patients with local recurrence are shown in the table below.

Case No. Recurrent Site Synchronous Metastasis

1 Pelvis Nil

2 Pelvis Liver

3 Abdominal wall Paraaortic lymph nodes

4 Pelvis Liver

5 Pelvis Liver

6 Abdominal wall Nil

7 Psoas muscle Liver

8 Abdominal wall Nil

9 Anastomosis Ovary

10 Bladder Nil

11 Anastomosis Lung

12 Abdominal wall Nil

13 Abdominal wall Carcinomatosis

Based on the clinical features we presented above, these cases could be distinguished from metachronous colon cancer.

Comments 5: 

In line 280, authors mentioned an analysis comparing the four chemotherapy regimens showed no significant differences in DFS among them. Can you provide the reference?

Response 5:

Yes. We thank the reviewer for the insightful comments. Univariate analysis of chemotherapy regimens for disease-free survival are provided in the table below and in the revised manuscript

Univariate analysis of chemotherapy regimens for disease-free survival

 No. of patients 3-year DFS rate p

Regimen 

FOLFOX / XELOX 108 76.3% 0.506

UFUR / Capecitabine 74 73.4% 

The sample sizes of those on XELOX and capecitabine were too small for conclusions. As a result, we compared between FOLFOX/XELOX versus UFUR/Capecitabine instead. (Results section, page 13, lines 203-206)

Comments 6: 

In this study, authors should define current smoker and former smoker more accurately. You can see a paper which was published at J Clin Oncol. 2013 Jun 1;31(16):2016-23. Especially former smoker, some patients might just stop smoking before operation. Another issue is previous smoking might already lead to lung injury, which may also lead to elevation of CEA.

Response 6:

Yes. We agree with the reviewer’s astute concern, some patients might just have stopped smoking before the operation, and previous smoking might have already led to lung injury. However, the lung function and CEA levels could significantly improve after 3 months of smoke cessation [1]. Thus, we placed former smokers in the group NS along with never smokers, rather than in the S group, which included patients who smoked both before and after the surgery. 

We agree that CEA levels are supposed to be different between never smokers and former smokers. However, one of the limitations of our study include the relatively small number of former smokers (n = 26), which likely resulted in limited statistical power. The amount and duration of smoking and type of cigarette would be taken into consideration in our future studies.

1. Pezzuto A, Spoto C, Vincenzi B, Tonini G. Short-term effectiveness of smoking-cessation treatment on respiratory function and CEA level. J Comp Eff Res. 2013;2: 335-343.

Reviewer #2

Comments 1: 

According to your statistic method, I don’t think it support your conclusion and the title. You should focus on the patients with persistent elevated CEA and using NS/S to compare disease-free survival if prognostic factors are balanced between two groups.

Response 1:

Yes. We thank the reviewer for the thoughtful comments. We did compare disease-free survival between smoker and nonsmoker patients with persistent elevated CEA (Results section page 12, in line 189). In subgroup analysis, smokers had higher 3 year-DFS rates than nonsmokers in group C (83.3% vs. 43.9%, p = 0.029; Figure 2). 

According to your suggestion, we have revised our title for the manuscript. Further, we also added more descriptions and two figures below and in the revised manuscript to demonstrate the 3 year-DFS rates between smokers and nonsmokers in different subgroups. 

“The overall 3-year DFS rate was similar between smokers and nonsmokers (80.2% vs. 77.1%, p = 0.485; Fig 2a). In the subgroup analysis, the 3-year DFS rate was also similar between smokers and nonsmokers in groups A and B (78.8% vs. 81.8%, p = 0.853; Fig 2b). In contrast, smokers had higher 3 year-DFS rate than nonsmokers in group C (83.3% vs. 43.9%, p = 0.029; Fig 2c).” (Results section, pages 12-13, lines 189-193)

Fig 2. Disease-free survival rates between smokers and nonsmokers (a) Overall disease-free survival (DFS) rates. (b) patients with normal postoperative serum CEA levels. (c) patients with elevated postoperative serum CEA levels.

Reviewer #3

Comments 1: 

In group C, the author defined elevated pre-operative and post-operative CEA group. However, in table1, the range of pre-operative CEA level is 2.8-263.5 ng/ml. Which means some patient’s CEA level was normal before operation. The author should carefully defined the group C.

Response 1:

Yes. We thank the reviewer for the thoughtful comments. There were 6 patients with normal preoperative CEA and elevated postoperative CEA in group C. One of them had tumor recurrence as carcinomatosis, 10 months after curative surgery. The clinical features of the patients are shown in the table below.

Case No. Preoperative CEA Postoperative CEA Recurrence

1 4.8 7.4 Carcinomatosis

2 4 6.1 Nil

3 2.8 6.5 Nil

4 3.7 9.3 Nil

5 3.9 6.4 Nil

6 4.1 5.1 Nil

According to your suggestion, we have redefined group C as patients with elevated postoperative CEA in the revised manuscript. 

Comments 2: 

One important finding in this study is post-operative CEA level is significantly higher in smoking group than non-smoking group (2.6 vs. 3.1 ng/ml). Although the p value is significant, the difference is too small in clinical practice. My other question is in non-smoking group, the mean post-operative CEA level is 2.6 ng/ml, range from 0.5-84.9 ng/ml. However, in smoking group, the mean post-operative CEA level is 3.1 ng/ml, range from 0.5-12.5 ng/ml. It seems the distribution of CEA level in non-smoking group is very wide. Do the author have the confidence to claim the post-operative CEA level is really higher in smoking group than non-smoking group?

Response 2:

Yes. We thank the reviewer for the thoughtful comments. We have used Mann-Whitney U test to compare the postoperative CEA between smokers and nonsmokers. Although there are few mild outliers and extreme outliers among the nonsmoker group, the difference in the median postoperative CEA between smokers and nonsmokers was statistically significant (p = 0.009). The distribution of postoperative CEA is shown in the box and whisker plot below.

Although the absolute median difference between the two groups is relatively small, the proportion of patients with elevated postoperative CEA is significantly higher in the smoker group (32% vs. 14%, p <0.001). However, smokers with elevated postoperative CEA levels had a higher 3 year-DFS rates than non-smoking patients with similar postoperative CEA levels (83.3% vs. 43.9%, p = 0.029). These findings support our conclusion that postoperative serum CEA levels showed no prognostic efficacy for smoker patients

Comments 3: 

In abstract, the post-operative CEA level in smoking group is 2.5 ng/ml. However, in the table 2, the level is 2.6 ng/ml.

Response 3:

Yes, we thank the reviewer for the kind reminder. We have corrected the abstract data in the revised manuscript.

---

## [Decision Letter · Decision Letter 1]

12 May 2020

Prognostic value of postoperative serum carcinoembryonic antigen levels in colorectal cancer patients who smoke

PONE-D-20-04276R1

Dear Dr. Huang,

We are pleased to inform you that your manuscript has been judged scientifically suitable for publication and will be formally accepted for publication once it complies with all outstanding technical requirements.

With kind regards,

Jason Chia-Hsun Hsieh, M.D. Ph.D

Academic Editor

PLOS ONE

Additional Editor Comments (optional):

All the questions were answered adequately.

Reviewers' comments:

Reviewer's Responses to Questions

**Comments to the Author**

1. If the authors have adequately addressed your comments raised in a previous round of review and you feel that this manuscript is now acceptable for publication, you may indicate that here to bypass the “Comments to the Author” section, enter your conflict of interest statement in the “Confidential to Editor” section, and submit your "Accept" recommendation.

Reviewer #1: All comments have been addressed

Reviewer #3: All comments have been addressed

2. Is the manuscript technically sound, and do the data support the conclusions?

Reviewer #1: Yes

Reviewer #3: Yes

3. Has the statistical analysis been performed appropriately and rigorously? 

Reviewer #1: Yes

Reviewer #3: Yes

4. Have the authors made all data underlying the findings in their manuscript fully available?

Reviewer #1: Yes

Reviewer #3: (No Response)

5. Is the manuscript presented in an intelligible fashion and written in standard English?

Reviewer #1: Yes

Reviewer #3: Yes

6. Review Comments to the Author

Reviewer #1: Thanks to the detailed and comprehensive response. I here have no further comments for this revised edition

Reviewer #3: (No Response)

7. PLOS authors have the option to publish the peer review history of their article (what does this mean?). If published, this will include your full peer review and any attached files.

Reviewer #1: No

Reviewer #3: No

---

## [Editor Report · Acceptance letter]

21 May 2020

PONE-D-20-04276R1 

Prognostic value of postoperative serum carcinoembryonic antigen levels in colorectal cancer patients who smoke 

Dear Dr. Huang:

I am pleased to inform you that your manuscript has been deemed suitable for publication in PLOS ONE. Congratulations! Your manuscript is now with our production department. 

With kind regards,

on behalf of

Dr. Jason Chia-Hsun Hsieh 

Academic Editor

PLOS ONE